# The Role of Type-2 Conventional Dendritic Cells in the Regulation of Tumor Immunity

**DOI:** 10.3390/cancers14081976

**Published:** 2022-04-13

**Authors:** Yasuyuki Saito, Satomi Komori, Takenori Kotani, Yoji Murata, Takashi Matozaki

**Affiliations:** Division of Molecular and Cellular Signaling, Department of Biochemistry and Molecular Biology, Kobe University Graduate School of Medicine, 7-5-1 Kusunoki-cho, Chuo-ku, Kobe 650-0017, Japan; skomori@med.kobe-u.ac.jp (S.K.); kotani@med.kobe-u.ac.jp (T.K.); ymurata@med.kobe-u.ac.jp (Y.M.); matozaki@med.kobe-u.ac.jp (T.M.)

**Keywords:** type-2 conventional dendritic cell, tumor immunity, antigen presentation, DC vaccine

## Abstract

**Simple Summary:**

Recent studies revealed that type-2 conventional dendritic cells (cDC2s) play an important role in antitumor immunity by promoting cytotoxic T-cell responses and helper T-cell differentiation. This review outlines the role of cDC2s in tumor immunity and summarizes the latest progress regarding their potential in cancer vaccination and cDC2-targeted cancer immunotherapy.

**Abstract:**

Conventional dendritic cells (cDCs) orchestrate immune responses to cancer and comprise two major subsets: type-1 cDCs (cDC1s) and type-2 cDCs (cDC2s). Compared with cDC1s, which are dedicated to the activation of CD8^+^ T cells, cDC2s are ontogenically and functionally heterogeneous, with their main function being the presentation of exogenous antigens to CD4^+^ T cells for the initiation of T helper cell differentiation. cDC1s play an important role in tumor-specific immune responses through cross-presentation of tumor-derived antigens for the priming of CD8^+^ T cells, whereas little is known of the role of cDC2s in tumor immunity. Recent studies have indicated that human cDC2s can be divided into at least two subsets and have implicated these cells in both anti- and pro-tumoral immune responses. Furthermore, the efficacy of cDC2-based vaccines as well as cDC2-targeted therapeutics has been demonstrated in both mouse models and human patients. Here we summarize current knowledge about the role of cDC2s in tumor immunity and address whether these cells are beneficial in the context of antitumor immune responses.

## 1. Introduction

Dendritic cells (DCs) are professional antigen-presenting cells that present endogenous and exogenous antigens to T cells and thereby orchestrate immunity or immune tolerance [1,2]. DCs are derived from blood-borne progenitors and localized in both lymphoid and nonlymphoid tissues. They consist of three major subsets termed plasmacytoid (pDCs), type-1 (cDC1s), and type-2 (cDC2s) conventional (or classical) DCs [3]. Monocytes also differentiate into monocyte-derived DCs (Mo-DCs) under inflammatory conditions [4]. DCs also reside in tumors, where they take up tumor-associated antigens (TAAs) and in turn migrate to tumor-draining lymph nodes in order to present these antigens to CD8^+^ or CD4^+^ T cells [5,6,7]. Antigen cross-presentation by cDC1s contributes to the priming of TAA-specific cytotoxic T cells [8]. cDC1s also support T helper 1 (Th1) cell polarization from naïve CD4^+^ T cells [9,10]. On the other hand, cDC2s are thought to comprise a heterogeneous population and preferentially prime naïve CD4^+^ T cells for Th2 or Th17 polarization [9,11,12,13]. Compared with other immune responses such as autoimmunity and infectious immunity, little is known about the role of the cDC2 subset in tumor immunity. In this review, we discuss the context-dependent functions of cDC2s in cancer immunology and consider the therapeutic potential of targeting these cells in cancer patients.

## 2. The Classification and Function of DCs in Mice and Humans

### 2.1. Mouse DC Lineages

cDCs were initially identified in mouse spleen, and they express integrin-αX (CD11c) and major histocompatibility complex (MHC) class II molecules at high levels [14,15]. In mice, cDC1s are characterized by cell-surface expression of XCR1 (a receptor for CXC chemokine XCL1), CD205 (DEC-205), and CLEC9A (DNGR1, a C-type lectin–like receptor) (Figure 1A) [16,17,18]. In secondary lymphoid tissues such as the spleen and lymph nodes, mouse cDC1s highly express CD8α on their surface, whereas nonlymphoid tissue–resident mouse cDC1s, including tumor-resident cells, express integrin-αE (CD103) [19]. cDC2s are characterized by cell-surface expression of signal regulatory protein α (SIRPα, also known as CD172a) and integrin-αM (CD11b) [20]. Mouse cDC1s and cDC2s develop from immediate cDC precursors termed pre-cDC1s and pre-cDC2s, respectively [21,22], both of which in turn are derived from DC lineage-restricted progenitors known as common DC progenitors (CDPs) [23,24,25]. The development of mouse cDC1s is dependent on the transcriptional factors IRF8 and BATF3 [26,27]. By contrast, the gene regulatory program that governs the differentiation of cDC2s is poorly defined, although IRF4, Notch2, RelB, and KLF4 are implicated in the development of lymphoid organ–resident cDC2s [15,28,29]. Mouse pDCs express CD11c at a low level and uniquely express B220, PDCA1, and SiglecH. The transcriptional factor E2-2 is required for the development of pDCs from DC progenitor cells, and IRF8 is also implicated in the differentiation of pDCs [30]. A recent study revealed that pDCs are also derived from common lymphoid progenitor cells (CLPs), and CLP-derived pDCs play a distinct role in type I interferon (IFN) production and antigen presentation compared with CDP-derived pDCs [31].

Various studies have shown that mouse cDC2s are heterogeneous and can be further divided into phenotypically and functionally distinct subsets. CD4^+^CD8α^−^ and CD4^−^CD8α^−^ (double-negative) subsets, which differ in their functional properties, are present in secondary lymphoid tissues [32,33]. Klf4^+^ or CD301b^+^ cDC2s have a unique potential to activate Th2 cells in lymphoid- and non-lymphoid organs [29,34]. cDC2s have also been subdivided on the basis of expression of the adhesion molecule ESAM [28]. ESAM^hi^ cDC2s express higher levels of MHC class II compared with ESAM^lo^ cDC2s, whereas ESAM^lo^ cDC2s show higher expression of colony-stimulating factor 1 receptor (CSF1R, also known as CD115), the chemokine receptor CCR2, lysozyme, CD14, and CD36, all of which are monocyte lineage markers [28]. Furthermore, mouse splenic cDC2s have recently been divided into subsets based on the transcriptional factors T-bet and RORγt [35]. T-bet^+^ cDC2s, termed cDC2As, were found to overlap phenotypically with ESAM^hi^ cDC2s, whereas RORγt^+^ cDC2Bs showed a similar phenotype to ESAM^lo^ cDC2s in that they expressed CSF1R and lysozyme. The cDC2A subset specifically expresses the transcriptional factor Nr4a3 and upregulates the expression of maturation markers such as CD83 and CCR7, whereas the cDC2B subset expresses CX3CR1 and CCR2 as well as the transcriptional factor C/EBPα [35]. Mo-DCs (or inflammatory DCs) share phenotypic features of both DCs and macrophages [36]. In mice, Mo-DCs share expression of CD11c and SIRPα with cDC2s but uniquely express FcγR1 (CD64) as well as the protein tyrosine kinase MerTK and CD88 [37,38,39].

### 2.2. Human DC Lineages

Both human cDC1s and cDC2s are thought to be evolutionarily conserved with the corresponding mouse cells, and they express CD11c and MHC class II. However, the molecules that specifically identify human cDC subsets differ in part from those in mice. In humans, cDC1s are identified by expression of CD141 (BDCA3) in addition to CLEC9A and XCR1 (Figure 1B) [40,41]. Human cDC2s are identified by preferential expression of CD1c (BDCA1) and CLEC10A (CD301a) in addition to SIRPα [41]. Human cDC subsets develop from bone marrow-derived progenitor cells. Human CDPs have the unique potential to give rise to DC lineage cells, with most of these progenitors having unipotent fates for pDCs, cDC1s, or cDC2s [42,43]. In peripheral blood, CD45RA^+^CD123^+^ cells include a precursor fraction for cDC1s and cDC2s [44]. Furthermore, HLA-DR^+^CD141^−^AXL^+^SIGLEC6^+^ cells include pre-cDCs that mainly differentiate into cDC2s [45]. BATF3 is also highly expressed in human cDC1s and critical for the generation of cDC1s in vitro [17]. Studies of individuals with heterozygous mutations of IRF8 revealed that IRF8 is required for the development of DC lineage cells, particularly for that of cDC1s [46]. Human pDCs uniquely express CD123 (IL-3R), CD303 (BDCA2), and CD304 (BDCA4). A recent study revealed that human pDCs can be subdivided into several populations in the context of antigen presentation and type I IFN production, particularly in response to pathogen stimulation [47].

Human cDC2s are also more phenotypically and functionally heterogeneous compared with other DC subsets (Table 1). Indeed, single-cell transcriptome analysis of human cDC subsets revealed that cDC2s can be divided into at least two distinct subsets designated DC2 and DC3 [45]. DC2s express CD5, whereas DC3s express CD14, CD36, and CD163 [45,48,49,50]. The DC2 population manifests the classical expression profile of CD1c^+^ cDCs, whereas the DC3 population appears closer to but distinct from monocytes or macrophages. Recent studies have also revealed that DC3s have an ontogeny profile distinct from that of DC2s [46,49].

Human Mo-DCs are present in the inflammatory milieu and are defined as CD14^+^CD1c^+^CD206^+^ cells. They secrete interleukin (IL)–17 and thereby induce the differentiation of naïve CD4^+^ T cells into Th17 cells [51].

## 3. Function of cDC2s

### 3.1. T Cell Activation

Mouse cDC1s cross-present exogenous or tumor-associated antigens to CD8α^+^ T cells through MHC class I molecules as well as secrete IL-12 [52]. cDC1s are thus thought to regulate cytotoxic T lymphocyte (CTL) and Th1 responses to intracellular pathogens and tumor cells [9,19]. Human cDC1s also have the capacity for antigen cross-presentation to CD8^+^ T cells as well as the capacity to promote the differentiation of Th1 cells [53]. In contrast, mouse cDC2s have very little cross-presentation capacity to CD8^+^ T cells, unless under specific circumstances upon stimulation with TLR7 agonist [54]. Human cDC2s are also capable of cross-presentation of antigens for priming of CD8^+^ T cells [55,56].

The function of cDC2s is largely restricted to antigen presentation by MHC class II to Th cells in secondary lymphoid organs in a subset-dependent manner [9]. For instance, Notch2-dependent ESAM^+^ mouse cDC2s were found to initiate Th17 cell differentiation by producing IL-23 and to play a protective role against *Citrobacter rodentium* infection [57], whereas mouse cDC2s dependent on the transcriptional factors IRF4 and KLF4 specifically contribute to Th2 cell differentiation [29,58]. As mentioned above, recently identified cDC2 subsets also have different capacity for T cell activation. The DC2B subset promotes differentiation of Th1 cells and Th17 cells, compared with the DC2A subset [35]. In addition to these lymphoid tissue–resident cDC2s, mouse migratory cDC2 subsets expressing CD301b are dedicated to Th2 priming during parasite infection [34]. Moreover, they induce the production of antigen-specific antibodies through activation of follicular Th cells [59]. Human cDC2s produce IL-12 and IL-23, which prime naïve T cells for development into Th1 and Th17 cells, respectively [60]. They also produce activin A and TGFβ, which are important for follicular Th cell differentiation [61]. In addition, human DC2s are prone to induce differentiation of naïve T cells into Th cells, particularly Th2 and Th17 cells, whereas human DC3s induce Th1 differentiation [50].

### 3.2. Toll-Like Receptor Expression and Cytokine Production by cDC2s

Proteomics analysis of cDC subsets has revealed that mouse cDC2s express Toll-like receptor (TLR) 7, which senses single-stranded RNA, as well as the viral recognition molecules RIG-I and MDA-5 [62]. These receptors are essential for type 1 IFN production in response to RNA virus infection [63]. TLR expression differs in a specific subset of mouse cDC2s. For example, the expression of TLR4, TLR8, and TLR9 is more prominent in the DC2B subset than in the DC2A [35]. Moreover, TLR7 and TLR8 expressions are upregulated in the human DC3 subset, whereas TLR3 expression is downregulated in the subset [50]. In addition, mouse splenic CD4^+^ cDC2s selectively produce tumor necrosis factor (TNF) receptor ligands such as TNF-α and lymphotoxin α_3_, which promote the proliferation and maintain the survival of fibroblastic reticular cells [64]. cDC2As mediate an anti-inflammatory response characterized by the upregulation of amphiregulin (Areg) and matrix metalloproteinase 9 (MMP9), which are thought to contribute to immune tolerance and tissue repair, respectively. By contrast, cDC2Bs produce larger amounts of proinflammatory cytokines such as TNF-α, IL-6, and IL-12 in response to their stimulation. Human CD1c^+^ cDC2s also produce various cytokines, such as IL-1β, IL-6, IL-12, and IL-23 [60,65]. In particular, DC3s are prone to secrete cytokines such as IL-1, TNF-α, IL-8, and IL-10 upon stimulation with TLR ligands [49,50].

## 4. The Role of cDC2s in the Regulation of Tumor Immunity

### 4.1. cDC2-Related Subsets within the Tumor Microenvironment

The tumor microenvironment (TME) contains both cDC1s and cDC2s [8,66,67,68,69], although they represent only a minor cell population compared with macrophages [8,67]. cDC1s play an important role in patrolling tumor tissue as well as in capturing and processing TAAs in the TME. They subsequently migrate to tdLNs, where they cross-present TAAs to CD8^+^ T cells for priming [70]. The role of cDC1s in tumor immunity has been reviewed in detail elsewhere [19,71,72]. By contrast, cDC2s present TAAs directly to CD4^+^ T cells or transfer them to lymphoid tissue–resident DCs, including other cDC2s [7,73]. Several studies have shown that CD4^+^ T cells are important for tumor immunosurveillance [74,75]. Human cDC2s also possess cross-priming activity for TAAs and CD8^+^ T cells in the TME [56]. cDC2s therefore have the potential to induce antitumor immunity mediated by CTLs and CD4^+^ effector T cells (Figure 2). Whereas the role of cDC2s in tumor immunity has been less well defined than that of cDC1s, in large part because of the paucity of specific markers to discriminate cDC2s from Mo-DCs, recent advances in methodologies such as single-cell RNA sequencing and multicolor flow cytometry have begun to reveal that cDC2s play an important role in such immunity.

Furthermore, mature types of DCs termed LAMP3^+^ DCs or “mature DCs enriched in immunoregulatory molecules” (mreg DCs) have recently been identified in mouse and human tumors [76,77]. Among these cells, mreg DCs were shown to be differentiated from cDC1s and cDC2s as a result of the uptake of TAAs and the upregulation of the expression of programmed cell death ligand 1 (PD-L1) and CD200 together with the activation markers CD40, CCR7, and IL-12β [77]. LAMP3^+^ DCs may also originate from cDC1s and cDC2s. They were shown to upregulate the expression of IDO1 and CCR7 and to have the potential for migration in tumors [76].

### 4.2. Pro-Tumoral Role of cDC2s

A tumor-protective role of cDC2s has been demonstrated in experimental mouse models. CD11b^+^ DCs that had infiltrated B16 melanoma can prime T cells but were shown to be characterized by reduced capacities for antigen uptake, antigen presentation, and migration to tdLNs, compared with normal skin DCs [78]. In addition, IRF4-dependent cDC2s were found to promote the growth of MC38 tumor cells and to inhibit the infiltration of Th1 and TNF-α–producing cells into the tumor [79]. These observations suggest that cDC2s, at least in part, attenuate antitumor immunity, possibly by limiting antitumor CD4^+^ effector T cell responses.

In humans, a myeloid cell population enriched in individuals with advanced cancer was shown to comprise cDC2s that coexpress CD1c and CD14 [49,51,80]. Moreover, cDC2s, Tregs, and exhausted T cells were enriched in lung cancer tissue, compared with normal lung tissue [81], suggesting that human cDC2s may induce immunosuppression and be associated with poor prognosis in human cancer. Of note, two-dimensional culture of human melanoma cells with DCs revealed that prostaglandin E_2_ and IL-6 released from the melanoma cells converted cDC2s into CD14^+^ cDCs, which are characterized by an immunosuppressive phenotype [82]. Such CD14^+^ cDCs were also characterized in patients with breast cancer, suggestive of a suppressive role in antitumor immunity [69,83]. Immunosuppressive factors in the TME may therefore alter the phenotype of cDC2s to one more closely resembling that of Mo-DCs, which may support tumor cell growth and result in a poor outcome in individuals with advanced cancer. Thus, the mechanisms possibly underlying the pro-tumoral action of CD14^+^ cDCs are likely to acquire the function of tumor-associate macrophages (TAM), which are thought to promote tumor cell growth. Indeed, tumor-infiltrated CD14^+^ cDCs express higher levels of TAM-related markers such as CD206, MerTK, and CD163 compared with blood-circulating CD14^+^ cDCs in melanoma patients [82].

### 4.3. Antitumor Response Mediated by cDC2s

A recent study showed that cDC2s are also associated with a good prognosis in patients with lung cancer [84]. In addition, enrichment of cDC2s in patients with lobular breast cancer, in particular estrogen receptor–positive breast cancer, was associated with favorable disease-free or overall survival [69,85]. Such improved survival associated with cDC2s was predominant within IFN-γ–dominant TME, suggesting that enrichment of cDC2s contributes to the infiltration of CTLs and M1 macrophages into the tumor [85].

The antitumor activity of cDC2s has also been experimentally demonstrated in a mouse melanoma model. An immunosuppressive relation between regulatory T cells (Tregs) and cDC2s in the TME or at the site of injection of irradiated granulocyte–macrophage colony-stimulating factor (GM-CSF)–producing B16 melanoma cells was identified [86]. Depletion of Tregs resulted in the upregulation of CCR7 expression in cDC2s and their migration to tdLNs, where they initiated differentiation of antitumor CD4^+^ Th cells. Moreover, it was recently reported that a tumor-infiltrated cDC2 subset represents genes upregulated by type I interferon stimulation. The activated cDC2 subset, named ISG^+^ DCs, activates CTLs and promotes anti-tumor immunity in the absence of cDC1s using a mouse regressing fibrosarcoma model [87].

Among human DC subsets, blood CD1c^+^ cDC2s are more efficient in engulfing and processing exogenous particulate antigens through phagocytosis than other DC subsets [88]. Moreover, human cDC2s have a potential to mediate anti-tumor immune responses induced by immunogenetic cancer cell death, a form of apoptosis characterized by the release of endogenous danger signals [89]. Human blood cDC2s also promote NK cell cytotoxicity in cooperating with pDCs [90]. Together, these various observations suggest that heterogeneity of cDC2s may underlie their apparent pro-tumorigenic and antitumorigenic roles in cancer. Further classification of cDC2s will therefore be required to provide a greater understanding of their role in cancer.

The DC3 subset of human cDC2s was recently found to preferentially accumulate in human papillomavirus (HPV)–associated oropharyngeal squamous cell carcinoma tissue [91]. DC3s produce proinflammatory cytokines including IL-12p70, IL-1β, GM-CSF, IL-6, and TNF-α, and they stimulate Th1 cells to mediate antitumor immune responses. Furthermore, DC3 infiltration in human breast cancers correlates with the abundance of CD8^+^ CD103^+^ CD69^+^ tissue–resident memory T cells, suggesting that DC3s initiate antitumor surveillance and immunity by activating resident memory T cells [49]. Thus, these findings have shown that cDC2 subsets infiltrate tumors, where they mature and regulate antitumor immune responses.

## 5. Targeting cDC2s for Cancer Immunotherapy

### 5.1. DC Vaccines

On the basis of their marked capacity for uptake and presentation of TAAs to T cells, DCs loaded with TAAs have been applied for cellular vaccination against tumors in preclinical and clinical studies. DCs generated from peripheral blood mononuclear cells have been administered as patient-derived DC vaccine formulations [92]. Such ex vivo–generated DCs have the potential to induce TAA-specific immunity, but they have induced only limited objective tumor regression in clinical studies [93], suggesting that subset specificity needs to be taken into account in the application of DC vaccines [94].

As an alternative to ex vivo–generated DCs, vaccination with naturally occurring DCs isolated from peripheral blood has been examined for the DCs’ ability to induce antitumor immune responses. cDC1s are present at approximately one-tenth the frequency of cDC2s in peripheral blood, but they have great potential for cross-presentation of TAAs to CTLs. Human cDC2s also have an intrinsic capacity for cross-presentation of TAAs to CTLs in addition to Th cell priming, and they are therefore also a potential source of naturally occurring DCs. Indeed, cDC2 vaccination was found to be more beneficial for the induction of antitumor immune responses, particularly those mediated by Th17 cells, than was cDC1 vaccination in mice [66] (Table 2). This antitumor effect of cDC2s was dependent on CD4^+^ T cells, whereas the contribution of CD8^+^ T and natural killer cells was found to be minimal. DC vaccines based on naturally occurring cDC2s have also been administered in patients with solid tumors. In addition, intra-tumoral injection of autologous CD1c^+^ cDCs along with immune checkpoint inhibitors such as ipilimumab and avelumab and intravenous injection of nivolumab induced early signs of antitumoral activity in some patients [95]. Vaccination with naturally circulating primary CD1c^+^ cDC2s was found to induce de novo immune responses and objective clinical responses even in patients with advanced metastatic melanoma [96]. In addition, isolated cDC2s pulsed with prostate TAAs increased the number of functional CTLs in tumor tissue and promoted antigen-specific humoral immune responses after long-term infusion [97,98]. Moreover, a cDC2 vaccine was found to promote the infiltration of CD4^+^ T cells into human melanoma tumors through expression of chemokines associated with Th2 and Th17 responses [99].

In addition, CD34^+^ cell-derived CD1c^+^ cDC2s exhibit a similar gene expression pattern to naturally isolated cDC2s and have a strong potential for generating antigen-specific CD8 T cell immunity [100,101]. Thus, CD34^+^ cell–derived cDC2-based vaccines will also be potent inducers of anti-tumor responses mediated by T cells.

### 5.2. cDC2-Targeted Therapy

#### 5.2.1. SIRPα

SIRPα is a transmembrane protein specifically expressed on myeloid lineage cells including DCs, monocytes–macrophages, and neutrophils. Among DCs, cDC2s and cDC1s express SIRPα at high and low levels, respectively [103,104]. SIRPα possesses an immunoreceptor tyrosine-based inhibition motif (ITIM) in its intracellular region and is thought to mediate inhibitory signaling in response to stimulation with its ligand CD47, which, like SIRPα, is a member of the immunoglobulin (Ig) superfamily of proteins. The NH_2_-terminal IgV-like domain of SIRPα interacts with the IgV-like domain of CD47. Studies with SIRPα- or CD47-deficient mice have shown that SIRPα is essential for homeostatic regulation of cDC2s in secondary lymphoid organs [104,105,106], and recent work suggested that chronic activation of the cDC2 subset was involved in such homeostatic regulation of the subset, at least in the spleen [107]. Studies of human tumor samples found that many tumor cell types highly express CD47, and that such expression is related to poor prognosis in cancer patients [108,109]. Treatment with antibodies to CD47 on tumor cells promoted antibody-dependent cellular phagocytosis of tumor cells mediated by macrophages [110]. In addition to the inhibition of macrophage-mediated phagocytosis, CD47 expressed on tumor cells was shown to inhibit the detection of tumor cell–derived mitochondrial DNA by SIRPα on DCs, resulting in attenuation of type I IFN production by DCs [111]. The blockade of CD47-mediated inhibitory signaling by antibodies to CD47 also promoted tumor cell uptake by DCs and enhanced antitumor immune responses through cross-priming of CTLs in mice [112]. Antitumor effects of DCs were also observed in a melanoma model in which mice were vaccinated with ovalbumin-loaded SIRPα-deficient DCs [102]. Moreover, treatment with antibodies to SIRPα or to CD47 promoted antigen cross-priming of CD8^+^ T cells by cDC2s [113] as well as enhanced the antitumor effect mediated by CTLs in a mouse syngeneic tumor model in vivo [114]. Collectively, these observations suggest that inhibition of SIRPα signaling by treatment with antibodies that target the CD47-SIRPα system results in the activation of cDC2s, which in turn promotes cross-priming of TAA-specific CTLs.

#### 5.2.2. CLEC4A4

CLEC4A4, also known as DC immunoreceptor 2 (DCIR2), is a C-type lectin receptor predominantly expressed on cDC2s [115]. It also contains an ITIM in its intracellular region and thus has the potential to mediate negative signaling. Delivery of tumor-specific antigens through a glycomimetic CLEC4A4 ligand into cDC2s upon stimulation with antibodies to CD40 and TLR3 ligands resulted in the priming of antigen-specific Th1 cells [115]. In humans, targeting of antigens to DCIR in vitro allows cross-presentation to CD8^+^ T cells by human DC subsets, including cDC2s [116]. Injection of chimeric anti-CLEC4A4 (DCIR2) antibody fused with OVA, which is allowed to deriver tumor antigen to cDC2s, was found to inhibit tumor cell growth in a syngeneic mouse B16F10-OVA melanoma model [117]. Targeting of cDC2s through CLEC4A4 and the consequent induction of CD8^+^ T cell-mediated antitumor immune responses is thus also a potential approach to cancer immunotherapy.

#### 5.2.3. Integrin α_M_β_2_ (CD11b/CD18)

Integrin α_M_β_2_ is known as CD11b/CD18 that is broadly expressed on myeloid-lineage cells, including cDC2s, Mo-DCs, monocytes/macrophages, and neutrophils. The adenylate cyclase toxin (CyaA) of *Bordetella pertussis* binds to integrin α_M_β_2_ [118]. In fact, CyaA binds both human and mouse CD11b^+^ DCs and induces maturation of the subset TLR4-dependent manner [119]. Furthermore, CyaA-induced antigen delivery to CD11b^+^ DCs promotes antigen-specific CTL responses, suggesting that integrin α_M_β_2_ on cDC2s can be a therapeutic target for TAA delivery through CyaA. GTL001 is a fusion of HPV-16 and HPV-18 E7 proteins with detoxified CyaA domain [120]. Topical treatment of GTL001 with TLR7/8 agonist eradicated HPV-16 E7–expressing cervical carcinoma in a mouse model [121]. However, no clinical difference was observed between therapy and placebo group in a phase II study of GTL001 with TLR7/8 agonist in women with HPV16 and/or 18 infection [122].

## 6. Conclusions

Experimental and clinical studies have revealed that not only cDC1s but also cDC2s contribute to the induction of antitumor immunity. cDC2s are more abundant in lymphoid tissues as well as in peripheral blood than are cDC1s, and they are a promising target for cancer immunotherapy, including the development of cancer vaccines. Given that cDC2s are a heterogeneous population and induce both proinflammatory and antiinflammatory responses, established methods to isolate specific cDC2 subsets will be required for the effective application of these cells to cancer immunotherapy.

## Figures and Tables

**Figure 1 cancers-14-01976-f001:**
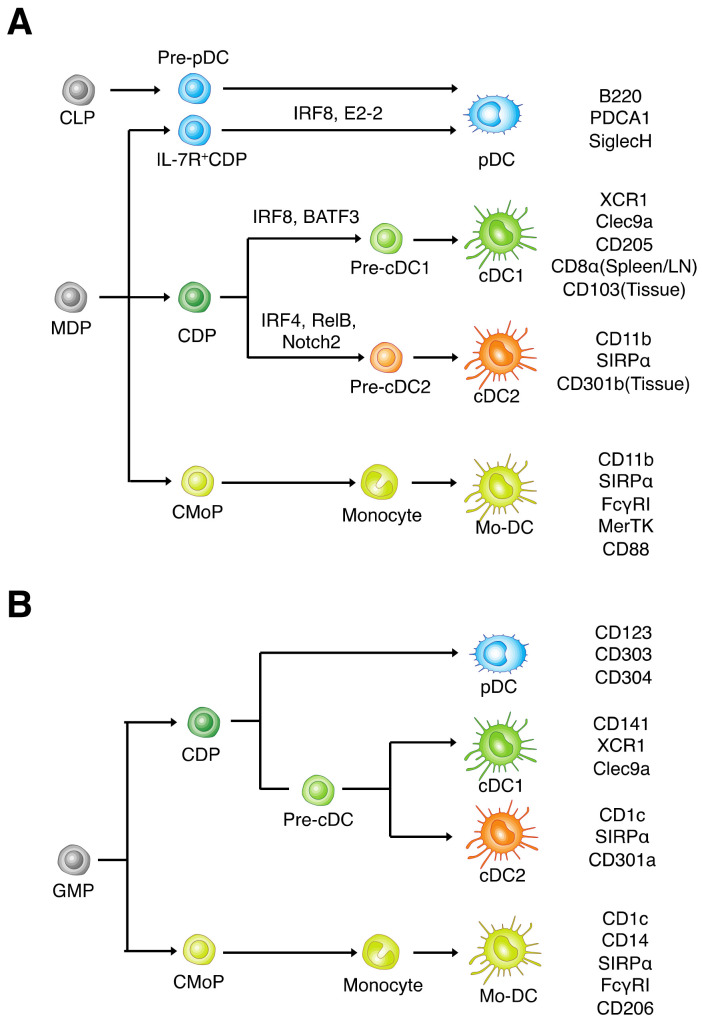
The development of mouse and human dendritic cell (DC) subsets. (**A**) Most mouse DCs are derived from macrophage–DC progenitors (MDPs), which give rise to DC lineage–restricted progenitors (CDPs). CDPs in turn give rise to precursors of cDC1s or cDC2s (pre-cDC1s and pre-cDC2s, respectively), which migrate from peripheral blood into lymphoid (LN, lymph node) and nonlymphoid tissues, where they undergo terminal differentiation into cDC1s or cDC2s. pDCs are derived from either DC (IL-7R^+^ CDP) or lymphoid (CLP) progenitors, and Mo-DCs differentiate from monocytes. Markers for each DC subset are shown on the right, and transcriptional factors that regulate the differentiation of each subset are indicated above the corresponding pathway. (**B**) Human DCs are derived from granulocyte-macrophage progenitors (GMPs), which give rise to CDPs with a developmental potential for differentiation into cDC1s, cDC2s, and pDCs. Mo-DCs differentiate from monocytes, which are derived from their progenitors (CMoPs). Markers for each DC subset are shown on the right, and transcriptional factors that regulate the differentiation of each subset are indicated.

**Figure 2 cancers-14-01976-f002:**
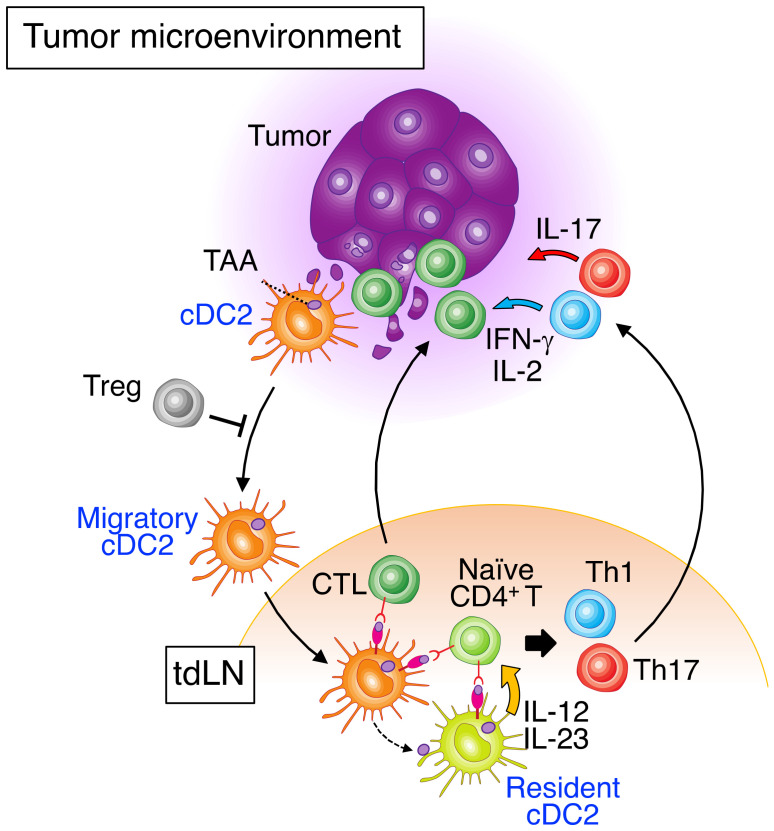
Overview of cDC2 functions in tumor immunity. Uptake of tumor-associated antigens (TAAs) triggers the activation of cDC2s characterized by the upregulation of costimulatory molecules on the cell surface and migration of the cells to tumor-draining lymph nodes (tdLNs). Regulatory T cells (Tregs) inhibit the migration of cDC2s. Migratory cDC2s present TAAs to naïve CD4^+^ T cells either directly or indirectly via transfer of TAAs to resident DC subsets including other cDC2s. These resident DCs also secrete IL-12 and IL-23, which trigger the differentiation of naïve CD4^+^ T cells into effector Th1 and Th17 cells, respectively. In addition, human cDC2s have a high intrinsic capacity for cross-presentation of TAAs to cytotoxic T lymphocytes (CTLs). CTLs attack tumor cells, and effector Th cells secrete IL-2, IFN-γ, and IL-17 and induce antitumor immune responses.

**Table 1 cancers-14-01976-t001:** Classification and function of human cDC2 subsets.

DC2			DC3			
Definition	TFs/Other Markers	Functional Properties	Definition	TFs/Other Markers	Functional Properties	Ref.
CD5^+^	IRF4	Increased chemotaxis toward CCL21 Induce IL-10^+^, IL-22^+^, IL-4^+^ T cell polarization	CD5^−^	MAFB/ S100A9	Induce IFN-γ^+^ T cell polarization	[50]
CD32B^+^		Higher MHC II expression	CD36^+^ CD163^+^	S100A9	Higher inflammatory genes expression	[45]
CD5^+^			CD14^+^ CD163^+^		Higher proinflammatory response	[48]
CD5^+^		Flt3-dependent Higher secretion of CCL5	CD14^+^ CD163^+^	S100A9	GM-CSF dependent Higher secretion of IL-1β, TNF-α, and CCL2 Induction of CD103^+^ T_RM_	[49]
CD5^+/−^ CD163^−^	IRF8^hi^/ BTLA	CD123^+^ GMP-dependent	CD14^+^/ CD163^+^ CD36^+^	IRF8^lo^	CD33^+^ GMP-dependent Higher secretion of IL-1β, TNF-α, and IL-10	[46]

Abbreviations: TF, transcriptional factor; Th, T helper; T_RM_, resident-memory T cell.

**Table 2 cancers-14-01976-t002:** cDC2 vaccines for cancer immunotherapy.

Species	Protocol	Tumor Type	Effects of DC Vaccine	Outcome	Ref.
Mouse	Sorted tumor-associated cDC2s injected 6 and 3 weeks before tumor cell injection	LLC-OVA	Th17 cell–dependent antitumor response CD4^+^ T cell–dependent antitumor response	Inhibition of tumor growth	[66]
	Syngeneic C57BL/6 mice injected with LV-miSIRPα–BMDCs pulsed with OVA peptide before tumor cell injection	EG7 or B16-OVA	Induction of IFN-γ–producing CD4^+^ and CD8^+^ T cells	Inhibition of tumor growth	[102]
Human	Isolated CD1c^+^ DCs from HLA-A*0201 patients were loaded with gp100 peptides in the presence of GM-CSF and then injected subcutaneously	Melanoma	High CD107a expression on CD8^+^ T cells IFN-γ, TNF-α, and CCL4 production by CD8^+^ T cells	Prolonged PFS in patients harboring tumor antigen-specific T cells	[96]
	Isolated CD1c^+^ DCs were loaded with gp100 peptides in the presence of GM-CSF and then injected subcutaneously	Melanoma	Infiltration of CD8^−^ and CD8^+^ T cells in tumor	Objective clinical responses	[99]
	CD1c^+^ DCs isolated from HLA-A*0201 patients were loaded with HLA-A*0201–restricted peptides and then injected intranodally	Prostate	Increased antigen-specific CD8^+^ T cells in the blood Increased antigen-specific humoral immune response	Prolonged PFS in patients harboring tumor antigen-specific T cells	[97]
	CD1c^+^ DCs isolated from HLA-A*0201 patients were loaded with HLA-A*0201–restricted peptides and then injected intradermally or intravenously	Prostate	Increased antigen-specific humoral immune response	Median survival of 18 months	[98]
	Intratumoral administration of unmanipulated CD1c^+^ DCs plus ipilimumab and avelumab in combination with intravenous low-dose nivolumab	Ovarian Breast Thyroid Melanoma	Infiltration of CD8^+^ T cells in tumor	Durable partial responses	[95]

Abbreviations: LLC-OVA, Lewis lung carcinoma expressing ovalbumin; Th17, T helper 17; LV-miSIRPα–BMDCs, bone marrow-derived DCs infected with a lentiviral vector for SIRPα microRNA; pDC, plasmacytoid DCs; PFS, progression-free survival.

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
