# Peer review of "The Role of Type-2 Conventional Dendritic Cells in the Regulation of Tumor Immunity"

_cancers, 2022, doi:10.3390/cancers14081976_

Round 1

Reviewer 1 Report

Major comments:

This is an interesting review on a timely topic. IRF4-dependent type 2 DCs are heterogeneous and their contribution to anti-tumor immunity is the topic of multiple investigations.

  • The authors should quote original paper (as much as possible) instead of reviews, especially in the introduction.
  • The several sections are not properly distributed. The DC classification in human and mouse, as well as their respective functions are well described. However, this part is over-sized as compared to the part describing specific functions of the cDC2 in anti-tumor responses and the therapies targeting them. Overall, the review would gain to be re-equilibrated in favor of the second part, id est, the role of cDC2 in cancer immunity.
  • The mechanisms possibly underlying the pro-tumoral action of DC2s or CD1c+CD14+ cells should be explained or at least discussed in more details.

Specific comments:

Abstract: “They consist of three major subsets termed plasmacytoid 31 (pDCs) as well as type 1 (cDC1s) and type 2 (cDC2s) conventional (or classical) DCs, all 32 of which are derived from DC lineage–restricted progenitors”. This sentence assume that the pDC are all differentiating from the CDP. The pDC have been shown to have two seperated progenitor, the Common Lymphoid Progenitor (CLP) and the CDP. This sentence should be modified to include the CLP as a pDC progenitor.

112: “CD123+ granulocyte-macrophage 112 progenitors (GMPs) have the unique potential to give rise to DC lineage cells, with most 113 of these progenitors having unipotent fates for pDCs, cDC1s, or cDC2s [35,36]. “ The CDP has been identified in human as well and represent a closer and more specific progenitor for the cDC1 and cDC2.

114. “In peripheral blood, CD45RA+CD123lo cells include a precursor fraction for cDC1s and cDC2s [37].“ This precursor in blood is CD45RA+CD123+

143. There are multiple evidence that bot murine and human DC2 can cross-present exogenous antigens by MHCI to CD8+ T cells, especially upon activation by PRRs. This should be quoted somewhere in the review.

149. Are KLF4-dependent DCs distinct from CD301b DCs? This should be discussed and data from Tussiwand et al. Immunity paper suggest that CD301b+ DCs are KLF4 dependent. This should be discussed.

170. The relationship between DC2B and DC3s is unclear (Cf Anderson and Murphy, review in 2020). Please explicit and quote the relevant literature for cytokine production by DC3s (Dutertre, 2019, Bourdely 2020, id est).

206. “4.1. Tumor Protective Role of cDC2s”: The authors could rephrase this title of section as “4.1 Pro-tumoral role of cDC2s”as a less ambiguous title.

208. “CD11b+ DCs that had infiltrated B16 melanoma were thus shown to be characterized by reduced capacities for antigen uptake, antigen presentation, and migration to tdLNs [70].” Even if the cDC2s are less capable to uptake; migrate to the tdLN and prime naïve T cells compared to the cDC1s, they have this capacity. This should be amended.

209 “In addition, cDC2s were found to inhibit the growth of MC38 tumor cells as well as the infiltration of Th1 and TNF-α–producing cells in a study of DC-specific Irf4 knockout mice, in which the number of cDC2s in secondary lymphoid organs was markedly reduced [71].

Even if in this study the abundance of cDC2 was not impaired in the IRF4ΔDC mice

These observations suggest that cDC2s attenuate antitumor immunity, possibly by limiting antitumor CD4+ effector T cell responses “ The study by Zhang, X et al. is showing the contrary at the level of MC38 growth. The cDC2 promotes MC38 tumor growth: in IRF4 cKO mice, MC38 are smaller.

222. Please quote Bourdely et al. 2020 as well evidencing CD14+CD1c+ in breast cancer as well.

227. Should mReg/LAMP3 DCs belong to paragraph about the “immunogenic side of DC2s”? Perhaps interesting to put in perspective the data from Maier & Merad et al. 2020, Nature.

290. This paper should be quoted: Yi et al., 2015, Immunity 43, 764–775

323. Here the mechanisms underlying the effect of CLEC4A4 ligand should be detailed. Does this refers to blockade or agonistic treatment?

Figure1: CD301b is mentioned in the Figure1 but not in the text. The authors should add CD301b in the text.

Author Response

Responses to Reviewer#1

Comments:

The authors should quote original paper (as much as possible) instead of reviews, especially in the introduction.

We agree with the reviewer’s suggestion and have replaced review papers with original ones as following, in the revised manuscript (Page 1, lines 34, 36, 38, 40).

The several sections are not properly distributed. The DC classification in human and mouse, as well as their respective functions are well described. However, this part is over-sized as compared to the part describing specific functions of the cDC2 in anti-tumor responses and the therapies targeting them. Overall, the review would gain to be re-equilibrated in favor of the second part, id est, the role of cDC2 in cancer immunity.

We agree with the reviewer’s suggestion. In the revised manuscript, we have reduced sections describing DC subsets. In particular, Table 1 was indeed confusing because of the mixture of several different classifications, each of which is phenotypically and functionally overlapped but not comprehensively identical. Thus, we deleted Table 1 in the revised manuscript. Instead, we have gained the volume of the latter part focusing on the role of cDC2 in cancer immunity.

The mechanisms possibly underlying the pro-tumoral action of DC2s or CD1c+CD14+ cells should be explained or at least discussed in more details.

According to the reviewer’s suggestion, we described the mechanisms possibly underlying the pro-tumoral role of cDC2s in section 4.2 (Page 8, lines 234-239). The mechanisms possibly underlying the pro-tumoral action of CD14+ cDCs are likely to acquire the function of tumor-associate macrophages (TAM), which are thought to pro-mote tumor cell growth. Indeed, tumor-infiltrated CD14+ cDCs express higher levels of TMA-related markers such as CD206, MerTK, and CD163 compared with blood circu-lating CD14+ cDCs in melanoma patients (Blasio et al. Nat Commun, 2020)

Specific comments:

  1. Abstract: “They consist of three major subsets termed plasmacytoid (pDCs) as well as type 1 (cDC1s) and type 2 (cDC2s) conventional (or classical) DCs, all of which are derived from DC lineage-restricted progenitors”. This sentence should be modified to include the CLP as a pDC progenitor.
    We think the reviewer is referring to the sentence in the introduction. Since the ontogeny of these DCs is described in the latter part of this section, we just remove the sentence “all of which are derived from DC lineage-restricted progenitors” from this section in the revised manuscript (Page 1, lines 31-32). 
  1. The CDP has been identified in human as well and represent a closer and more specific progenitor for the cDC1 and cDC2.
    We agree with the reviewer’s comment. CD123+ GMP was identified as a subset having CDP potential at the clonal level. We replaced the term “CD123+ granulocyte-macrophage progenitors (GMPs)” with human CDP in the revised manuscript (Page 4, line 110).
  1. “In peripheral blood, CD45RA+CD123lo cells include a precursor fraction for cDC1s and cDC2s [37]. “This precursor in blood is CD45RA+CD123+
    According to the reviewer’s suggestion, we described “CD45RA+CD123+” cells as a precursor fraction for cDC1s and cDC2s in the revised manuscript (page 4, line 112).
  1. There are multiple evidence that both murine and human DC2 can cross-present exogenous antigens by MHCI to CD8+ T cells, especially upon activation by PRRs. This should be quoted somewhere in the review.
    According to the reviewer’s suggestion, we add the sentence “mouse cDC2s have very little cross-presentation capacity to CD8+ T cells, unless under specific circumstances upon stimulation with TLR7 agonist.” in the revised manuscript (page 5, lines 141-144).
  1. Are KLF4-dependent DCs distinct from CD301b DCs? This should be discussed and data from Tussiwand et al. Immunity paper suggest that CD301b+ DCs are KLF4 dependent. This should be discussed.
    We agree with the reviewer’s comment. We described that “Klf4 or CD301b cDC2s have a unique potential to activate Th2 cells in lymphoid- and non-lymphoid organs”, in the revised manuscript (Page 3, line 85 and page 4, line 86).

  2. The relationship between DC2B and DC3s is unclear. Please explicit and quote the relevant literature for cytokine production by DC3s.
    According to the reviewer’s suggestion, we added studies characterizing cytokine production from DC3s in the revised manuscript (page 6, lines 177-178).
  1. The authors could rephrase this title of section as “4.1 Pro-tumoral role of cDC2s” as a less ambiguous title.
    According to the reviewer’s suggestion, we change the title of the section to “Pro-tumoral role of cDC2s” in section 4.1 (Page 7, lines 214).
  1. “CD11b+ DCs that had infiltrated B16 melanoma were thus shown to be characterized by reduced capacities for antigen uptake, antigen presentation, and migration to tdLNs.” Even if the cDC2s are less capable to uptake; migrate to the tdLN and prime naïve T cells compared to the cDC1s, they have this capacity. This should be amended.
    According to the reviewer’s suggestion, we rewrote the result “CD11b+ DCs that had infiltrated B16 melanoma can prime T cells but were shown to be characterized by reduced capacities for antigen uptake, antigen presentation, and migration to tdLNs, compared with normal skin DCs”, in the revised manuscript (Page 7, lines 216-218)

  2. The study by Zhang, X et al. is showing the contrary at the level of MC38 growth. The cDC2 promotes MC38 tumor growth: in IRF4 cKO mice, MC38 are smaller.
    We appreciate the reviewer’s comment. As the reviewer pointed out, the study by Zhang et al showed that the number of CD11b+ DCs was not reduced in CD11c-Cre:Irf4flox mice, although they inhibited tumor cell growth. We thus rephrase the sentence as “Irf4-dependent cDC2s were found to promote the growth of MC38 tumor cells and to inhibit the infiltration of Th1 and TNF-α–producing cells into the tumor” in the revised manuscript (Page 7, lines 218-222).
  1. Please quote Bourdely et al. 2020 as well evidencing CD14+CD1c+ in breast cancer as well.
    According to the reviewer’s suggestion, we quoted the paper in the revised manuscript (Page 7, line 224).
  1. Should mReg/LAMP3 DCs belong to paragraph about the "immunogenic side of DC2s"? Perhaps interesting to put in perspective the data from Maier & Merad et al. 2020, Nature.
    We agree with the reviewer’s comment. We moved this part to section 4.1. “Phenotype of cDC2 with the tumor microenvironment” in the revised manuscript (Page 6, lines 196-203).
  1. This paper should be quoted: Yi et al., 2015, Immunity 43, 764–775
    According to the reviewer’s suggestion, we quoted the paper by Yi et al. in the revised manuscript (page10, lines 324-325).
  1. Here the mechanisms underlying the effect of CLEC4A4 ligand should be detailed. Does this refers to blockade or agonistic treatment?
    We applicate the reviewer’s comment. The term “CLEC4A4 ligand” is inappropriate to explain the study. In this study, the authors tested the effect of chimeric anti-CLEC4A4 (DCIR2) antibody fused with OVA, which is allowed to deliver tumor antigen to cDC2s. We rewrote the description of the study in the revised manuscript (Page 10, lines 349-353).
  1. The authors should add CD301b in the text.
    As replied in comment 5, we described the CD301b subset in the revised manuscript (Page 3, line 85, and page 4, line 86).

Reviewer 2 Report

Summary:

This review article by Saito et al., outlines the role of cDC2s in tumour immunity and summarizes the latest progress regarding their potential in cancer vaccination and cDC2-targeted cancer immunotherapy. They first provide a thorough review of the numerous subsets of cDC2 present in mice and humans. Then explain the role of cDC2 in activating T cells and briefly detail the cytokines they produce. Next the authors focus on the evidence for pro- and anti-tumorigenic functions of cDC2s before covering DC-vaccine approaches and some DC-targeted immunotherapies.

General comments:

In general, the description of cDC2 subsets in human and mice is quite extensive and well researched, being a good general review of the field. The following sections on DC function and role in tumour immunity could use some improvement to enhance the flow and focus of the review, but this can likely be addressed with some re-writing.

However, the review should be strengthened by spending more time developing the sections relating to DC-specific approaches in both the DC vaccination section (4.1) and DC-targeted therapies section (4.2). I do not think the breadth of work within the field has been sufficiently covered in these sections and therefore more work needs to be done to address this.

General review of each section:

Section 2. Classification and Function of DCs in Mouse and Human. This section is of acceptable quality and depth but is really just an introduction to the main topic of the review.

Section 3: Functions of cDC2s. This section could be improved through better integration of the functional roles of cDC2s within their biological niche and the suite of cytokines and chemokines which they produce to drive unique CD4+ T cell responses, support CD8+ T cell immunity and remodel their microenvironment.

Section 4: Role of cDC2s in Regulation of Tumor Immunity. This is a complex section and could be better integrated, the phenotype of cDC2 within the TME could be examined first to set the scene for their reported pro- and anti-tumour effects, to avoid the issue with discussing pro-tumourigenic effects versus simple DC dysfunction. Clear separation of human and mouse studies, where reported, would be useful throughout the review.

Section 5: Targeting cDC2s for Cancer Immunotherapy. The article summary states: the review outlines the role of cDC2s in tumor immunity and summarizes the latest progress regarding their potential in cancer vaccination and cDC2-targeted cancer immunotherapy. Therefore, this section should be a significant (and the most interesting) part of the review, fully updating the latest progress on cDC2-targeted therapies. I don’t think that the review in its current state fulfills this, and more work needs to be done reviewing the latest progress regarding cDC2-targeted therapies.

Specific comments:

  1. First, the review presents a good summary of the current nomenclature for classifying DC subsets in human and mouse across a broad range of tissues and their development from lineage-specific progenitors. However, there are some exceptions. For example, CD11b/CD103 double positive DCs are present in mouse mLN, a cDC2 subset which drives IL-17 production by CD4 T cells via production of IL-6 and IL-23 (Persson et al., 2013, Schlitzer et al., 2013). These are not covered in the Table 1, unless through overlap with another listed subset.

The Table 1 is somewhat confusing in this regard as multiple subsets are presented which likely overlap phenotypically and functionally, although this indeed reflects the state of the current literature on this subject, so is not an inaccurate representation.

  1. Next the review briefly touches on the role of cDC2 in activation of T cells (Section 3.1), with specific examples highlighting the specialisation of unique cDC2 subsets. This section could be examined in more detail to tease out the differences between subsets and properly highlight the complexity in cDC2 functions, with links made between this section and the following section on cytokine production, which are intrinsically linked. Also, the function of human cDC2, are supported by reference [51], which likely misses some of the specific complexity of the human cDC2 family.

Next, Section 3.2 examines cDC2 cytokine production. Here the authors single out TLR7 expression by cDC2, but TLR expression by cDC2 subsets in more extensive that just TLR7 and is often subset or tissue specific. More examination of TLR expression by cDC2 could be useful and highly relevant to later sections of the review.

  1. The next section examines role of cDC2 in regulating anti-tumour immunity, examining evidence for pro- and anti-tumourigenic effects.

At the beginning of Section 4. Lines 174-183 suffers from some repetition and could be re-written to improve clarity (see below A). Also lines 185-187 on cytokines should be in Section 3.2, which is specifically about cytokine production in cDC2 (see below B).

  1. cDCs have a strong capacity for migration to the tumor-draining lymph nodes (tdLNs), 176 where they present TAAs to T cells and thereby induce antitumor adaptive immune 177 responses [61]. cDC1s play an important role in patrolling tumor tissue as well as in 178 capturing and processing TAAs in the TME. They subsequently migrate to tdLNs, where 179 they cross-present TAAs to CD8+ T cells for priming [62]. The role of cDC1s in tumor 180 immunity has been reviewed in detail elsewhere [13,63,64]. 181 Likewise, cDC2s take up TAAs and migrate to tdLNs, where they present TAAs 182 directly to CD4+ T cells or transfer them to lymphoid tissue–resident DCs, including other 183 cDC2s [65,66].

  1. Furthermore, cDC2s produce a wide variety of cytokines 185 such as IL-1β, IL-6, IL-12, and IL-23 that are important for T cell activation and multiple 186 Th cell differentiation [61],

The final sentence of the paragraph is also confusing, Line 194- “In the latter part of this review, we discuss the specific contribution of cDC2s to tumour immunity”. Rather this done in the following sections 4.1/4.2 and the latter parts focus on DC vaccines and targeted therapies.

Section 4.1 is titled “Tumour protective role of cDC2s”, however it is unclear from the text whether cDC2 function is simply reduced by the suppressive TME or actively promotes/protects tumours. Evidence is given for both scenarios, this could be examined further or the phenotype of cDC2s in tumours discussed separately.

The authors suggest on Line 213- “cDC2s attenuate antitumor immunity, possibly by limiting antitumor CD4+ effector T cell responses”, but this into always the case. There are many reports of anti-tumour Th1, Th17 and Th9 CD4+ T cells, which are reliant on cDC2 for their generation, driving tumour control (Purwar et al., 2012, Lu et al., 2018, Muranski et al, 2008, Martin-Orozco et al., 2009) and the reality is likely to be highly complex.

The following section (4.2) should focus more on the role of cDC2 to drive anti-tumour immunity to reflect this. Instead, a single reference [78], looking specifically at the interaction between Treg and cDC2s and reporting clinical correlates is provided as evidence. This is a weakness of the review in some sections.

While mreg DCs are likely an important ‘group’ of DCs in anti-tumour immunity, as the author states on Line 250-they can be differentiated from cDC1 and cDC2. Therefore, it is difficult to prescribe their function specifically to cDC1 or cDC2 DCs, therefore their addition here adds ambiguity to the discussion rather than clarity.

  1. The final section focuses on targeting cDC2 for Cancer Immunotherapy. These sections need to focus specifically on reviewing the literature reporting these uses, providing less general discussion and more specific examples to strengthen the review.

Line 270 - Human cDC2s also have an intrinsic capacity for cross-presentation of TAAs to CTLs, and they are therefore also a potential source of naturally occurring DCs. It is not clear of what they are specifically a source for? A vaccine targeting CD4 T cell subsets or for cross-presentation to CD8 T cells?

Line 275/6 - By contrast, cDC2 vaccination also activated TAA-specific CTLs, and pDCs synergize such cross-priming ability of cDC2s in mouse lymphoma model [85]. The predominant view in the literature, at least for murine DC subsets, is that cDC2 do very little cross-presentation, unless under specific circumstances, this is normally restricted to cDC1.

Line 283 - Moreover, a cDC2 vaccine was found to promote the infiltration of Th2 cells into human melanoma tumors through expression of chemokines associated with Th2 responses [89]. What was the outcome of this study? As Th2 cells are often cited as being pro-tumourigenic as compared to Th1 cells?

In, Section 4.2 (cDC2-targeted therapies), I would argue that treating with a SIRPa antibody is not specifically designed to target cDC2, but rather has an important bystander effects on these cells. The section on Clec4a4 is indeed cDC2-specific therapeutic approach and of great interest to the audience. More focus on novel cDC2-targeted therapies e.g., novel cDC2-targeted vaccination strategies, specific use of TLRs targeting cDC2, chemokine therapy or targeting immune checkpoints or costimulatory markers on DCs (eg. VISTA, TIM-3, PD-L2) would greatly strengthen the review article.

Author Response

Responses to Reviewer#2

General comments:

In general, the description of cDC2 subsets in human and mice is quite extensive and well researched, being a good general review of the field. The following sections on DC function and role in tumour immunity could use some improvement to enhance the flow and focus of the review, but this can likely be addressed with some re-writing.

However, the review should be strengthened by spending more time developing the sections relating to DC-specific approaches in both the DC vaccination section (4.1) and DC-targeted therapies section (4.2). I do not think the breadth of work within the field has been sufficiently covered in these sections and therefore more work needs to be done to address this.

We appreciate the reviewer’s comment. We have gained the volume of the latter part focusing on the role of cDC2 in cancer immunity in both sections 4 and 5, in the revised manuscript.

Specific comments:

  1. CD11b/CD103 double positive DCs are present in mouse mLN, a cDC2 subset which drives IL-17 production by CD4 T cells via production of IL-6 and IL-23 (Persson et al., 2013, Schlitzer et al., 2013). These are not covered in the Table 1, unless through overlap with another listed subset
  2. The Table 1 is somewhat confusing in this regard as multiple subsets are presented which likely overlap phenotypically and functionally, although this indeed reflects the state of the current literature on this subject, so is not an inaccurate representation.
    We agree with the reviewer’s comment. Table 1 is indeed confusing because of the mixture of several different classifications, each of which is phenotypically and functionally overlapped but not comprehensively identical. Thus, we have deleted Table 1 in the revised manuscript.

  1. This section (3.1) could be examined in more detail to tease out the differences between subsets and properly highlight the complexity in cDC2 functions, with links made between this section and the following section on cytokine production, which are intrinsically linked. Also, the function of human cDC2, are supported by reference [51], which likely misses some of the specific complexity of the human cDC2 family.
    We agree with the reviewer’s comment. We did not specify the role of cDC2 subsets in the context of T cell differentiation and activation in this section in the initially submitted manuscript. We added several sentences mentioning the subset-specific role of DC2 in T cell differentiation and activation in the revised manuscript.

    Page 5, lines 150-152: “As mentioned above, recently identified cDC2 subsets also different capacity for T cell activation. DC2B subset promotes differentiation of Th1 cells and Th17 cells, compared with DC2A subset.”

    Page 5, lines 158-160: “In addition, human DC2s are prone to induce differentiation of naïve T cells into Th cells, particularly Th2 and Th17 cells, whereas human DC3s induce Th1 differentiation. “

  2. More examination of TLR expression by cDC2 could be useful and highly relevant to later sections of the review.
    According to the reviewer’s suggestion, we described TLR expression by cDC2 subsets in both human and mouse in the revised manuscript (page 6, lines 165-168). We also changed the title of the section to “Toll-like receptor expression and cytokine Production by cDC2s”, in the revised manuscript (page 5, line 161).

  1. A) At the beginning of Section 4. Lines 174-183 suffers from some repetition and could be re-written to improve clarity.
    According to the reviewer’s comment, we rewrote these sentences to avoid repetition in the revised manuscript on page 6, lines 183-188. “cDC1s play an important role in patrolling tumor tissue as well as in capturing and processing TAAs in the TME. They subsequently migrate to tdLNs, where they cross-present TAAs to CD8+ T cells for priming [70]. The role of cDC1s in tumor immunity has been reviewed in detail elsewhere [19,71,72]. By contrast, cDC2s present TAAs directly to CD4+ T cells or transfer them to lymphoid tissue–resident DCs, including other cDC2s [7,73].”

    B) Also lines 185-187 on cytokines should be in Section 3.2, which is specifically about cytokine production in cDC2.
    According to the reviewer’s suggestion, we move the sentence to Section 3.2 and delete some repetition phrases in the revised manuscript (Page 6, lines 175-178). 
  1. The final sentence of the paragraph is also confusing, Line 194- “In the latter part of this review, we discuss the specific contribution of cDC2s to tumour immunity”.
    We agree with the reviewer’s comment. We deleted the sentence in the revised manuscript.
  1. Section 4.1 is titled “Tumour protective role of cDC2s”, however it is unclear from the text whether cDC2 function is simply reduced by the suppressive TME or actively promotes/protects tumours. Evidence is given for both scenarios, this could be examined further or the phenotype of cDC2s in tumours discussed separately.
    According to the reviewer’s suggestion, we described the mechanisms possibly underlying the pro-tumoral action of DC2s, particularly CD1c+CD14+ cDCs, in the revised manuscript (Page 8, lines 234-239). They are likely to acquire the function of tumor-associate macrophages (TAM), which are thought to actively promote tumor cell growth. Indeed, tumor-infiltrated CD1c+CD14+ cDC2s express higher levels of TMA-related markers such as CD206, MerTK, and CD163 compared with blood circulating CD1c+CD14+ cDC2s in melanoma patients. We also change the title of the section to “Pro-tumoral role of cDC2s” (Page 7, line 214).
  2. There are many reports of anti-tumour Th1, Th17 and Th9 CD4+ T cells, which are reliant on cDC2 for their generation, driving tumour control and the reality is likely to be highly complex.
    We agree with the points suggested by the reviewer. We have toned down the description in this section and added the role of the Th cell in the anti-tumor response in section 4.2 in the revised manuscript (Page 7, lines 220-222).
  3. The following section (4.2) should focus more on the role of cDC2 to drive anti-tumour immunity to reflect this. Instead, a single reference [78], looking specifically at the interaction between Treg and cDC2s and reporting clinical correlates is provided as evidence. This is a weakness of the review in some sections.
    We agree with the reviewer’s comment. We added studies related to antitumor response mediated by cDC2s in the revised manuscript (Page 8, lines 253-262). “Moreover, it was recently reported that a tumor-infiltrated cDC2 subset represents genes upregulated by type-I interferon stimulation. The activated cDC2 subset, named ISG+ DCs, activates CTLs and promotes anti-tumor immunity in the absence of cDC1s using a mouse regressing fibrosarcoma model (Duong et al., Immunity, 2022). Among human DC subsets, blood CD1c+ cDC2s are more efficient in engulfing and processing exogenous particulate antigens through phagocytosis than other DC subsets [88]. Moreover, human cDC2s have a potential to mediate anti-tumor immune responses induced by immunogenetic cancer cell death, a form of apoptosis character-ized by the release of endogenous danger signals (Blasio et al., Oncoimmunology, 2016). Human blood cDC2s also pro-mote NK cell cytotoxicity in cooperating with pDCs (Beek et al., Oncoimmunology, 2016).”
  1. While mreg DCs are likely an important ‘group’ of DCs in anti-tumour immunity, as the author states on Line 250-they can be differentiated from cDC1 and cDC2. Therefore, it is difficult to prescribe their function specifically to cDC1 or cDC2 DCs, therefore their addition here adds ambiguity to the discussion rather than clarity.
    We agree with the reviewer’s comment. These are specific DC subsets found in the TME, but the origin of these subsets is not specific to cDC2s. We moved this part to new section 4.1. “cDC2-related subsets within the tumor microenvironment” in the revised manuscript (Page 6, lines 196-203).
  1. Human cDC2s also have an intrinsic capacity for cross-presentation of TAAs to CTLs, and they are therefore also a potential source of naturally occurring DCs. It is not clear of what they are specifically a source for? A vaccine targeting CD4 T cell subsets or for cross-presentation to CD8 T cells?
    We think naturally occurring DC2s are the source for both Th priming and cross-presentation to CTLs, based on the studies using these DCs. We rewrote this sentence in the revised manuscript (Page 9, line 289).
  1. By contrast, cDC2 vaccination also activated TAA-specific CTLs, and pDCs synergize such cross-priming ability of cDC2s in mouse lymphoma model [85]. The predominant view in the literature, at least for murine DC subsets, is that cDC2 do very little cross-presentation, unless under specific circumstances, this is normally restricted to cDC1.
    We agree with the reviewer’s comment. In this study, the contribution of murine cDC2 alone in the context of the anti-tumor effect is minimal. We have deleted the description of this study in the revised manuscript because it is inconsistent with the low cross-presentation capacity of murine cDC2s.

  2. Moreover, a cDC2 vaccine was found to promote the infiltration of Th2 cells into human melanoma tumors through expression of chemokines associated with Th2 responses [89]. What was the outcome of this study? As Th2 cells are often cited as being pro-tumourigenic as compared to Th1 cells?
    We appreciate the reviewer’s comment. We apologize that this sentence was a kind of overstatement. The study demonstrated the infiltration of CD4 and CD8, as well as NK cells in the skin, although there are variations between patient samples. Instead, ex vivo stimulated cDC2s from blood to secrete higher levels of chemokines such as CCL17 and CCL20, which are thought to be important for recruiting Th2 and Th17 cells. We rewrote this sentence highlighting the recruitment of CD4 T cells, not specific to Th2 cells, in the revised manuscript (Page 9, lines 303-305). In addition, this clinical study reported objective clinical responses in some patients (Table 2).
  1. In, Section 4.2 (cDC2-targeted therapies), I would argue that treating with a SIRPa antibody is not specifically designed to target cDC2, but rather has an important bystander effects on these cells. The section on Clec4a4 is indeed cDC2-specific therapeutic approach and of great interest to the audience. More focus on novel cDC2-targeted therapies e.g., novel cDC2-targeted vaccination strategies, specific use of TLRs targeting cDC2, chemokine therapy or targeting immune checkpoints or costimulatory markers on DCs (eg. VISTA, TIM-3, PD-L2) would greatly strengthen the review article.
    As the reviewer suggested, we added integrin αMβ2 (CD11b/CD18) as one of additional cDC2-targeted therapies. Integrin αMβ2 is known as CD11b/CD18, that is broadly expressed on myeloid-lineage cells, including cDC2s, Mo-DCs, Monocytes/Macrophages, and Neutrophils. The adenylate cyclase toxin (CyaA) of Bordetella pertussis binds to integrin αMβ2 (Guermonprez et al., J Exp Med, 2001). In fact, CyaA binds both human and mouse CD11b+ DCs and induces maturation of the subset in TLR4-dependent manner (Dadaglio et al., J Immunol, 2014). Furthermore, CyaA-induced antigen delivery to CD11b+ DCs promotes antigen-specific CTL responses, suggesting that integrin αMβ2 on cDC2s can be a therapeutic target for TAA delivery through CyaA. GTL001 is a fusion of HPV-16 and HPV-18 E7 proteins with detoxified CyaA domain (Esquerre et al., Plos One, 2017). Topical treatment of GTL001 with TLR7/8 agonist eradicated HPV-16 E7 expressing cervical carcinoma in mouse model (Esquerre et al., Vaccines, 2017). However, no clinical difference was observed between the therapy and placebo group in a phase II study of GTL001 with TLR7/8 agonist in women with HPV16 and/or 18 infection (Boilesen et al., Vaccines, 2021).
    With regard to other molecules the reviewer suggested, we could not find the specific role of cDC2s in the context of tumor immunity. For instance, in the tumor microenvironment, TIM-3 is preferentially expressed on CD103+ cDC1s but not cDC2s. V-domain Ig Suppressor of T cell Activation (VISTA) is a programmed death protein-1 (PD-1) homolog expressed on T cells and antigen-presenting cells, but the specific role of VISTA on cDC2s remained to be unclear. Furthermore, PD-L2 is expressed by both various immune cells and tumor cells, depending on microenvironmental stimuli. However, the role of PD-L2 on cDC2s in tumor immunity has not been clearly demonstrated yet.

Round 2

Reviewer 1 Report

The reviewer congratulates the authors for their improved version of the review. 

Reviewer 2 Report

I thank the authors for taking the time to address my feedback.